# Opicapone Improves Global Non-Motor Symptoms Burden in Parkinson’s Disease: An Open-Label Prospective Study

**DOI:** 10.3390/brainsci12030383

**Published:** 2022-03-12

**Authors:** Diego Santos García, Gustavo Fernández Pajarín, Juan Manuel Oropesa-Ruiz, Francisco Escamilla Sevilla, Raúl Rashid Abdul Rahim López, José Guillermo Muñoz Enríquez

**Affiliations:** 1CHUAC (Complejo Hospitalario Universitario de A Coruña), 15006 A Coruña, Spain; 2Hospital San Rafael, 15009 A Coruña, Spain; 3CHUS (Complejo Hospitalario Universitario de Santiago), Santiago de Compostela, 15706 A Coruña, Spain; gferpaj@gmail.com; 4Memory Unit, Hospital Universitario Juan Ramón Jiménez, 21005 Huelva, Spain; juaororui@gmail.com; 5Unidad de Trastornos del Movimiento, Servicio de Neurología, Hospital Universitario Virgen de las Nieves, Instituto de Investigación Biosanitaria (ibs.Granada), 18013 Granada, Spain; fescamilla@hotmail.com; 6Neurology Department, Puerta del Mar University Hospital, 11009 Cádiz, Spain; raulrashidlo@gmail.com; 7Complejo Público Asistencial de Zamora, 49022 Zamora, Spain; jgmeaadh6@hotmail.com

**Keywords:** effectiveness, non-motor symptoms, open-label study, opicapone, Parkinson’s disease

## Abstract

Patients with Parkinson’s disease (PD) can improve some non-motor symptoms (NMS) after starting treatment with opicapone. The aim of this study was to analyze the effectiveness of opicapone on global NMS burden in PD. OPEN-PD (Opicapone Effectiveness on Non-motor symptoms in Parkinson’s Disease) is a prospective open-label single-arm study conducted in 5 centers from Spain. The primary efficacy outcome was the change from baseline (V0) to the end of the observational period (6 months ± 30 days) (V2) in the Non-Motor Symptoms Scale (NMSS) total score. Different scales were used for analyzing the change in motor, NMS, quality of life (QoL), and disability. Thirty-three patients were included between JUL/2019 and JUN/2021 (age 63.3 ± 7.91; 60.6% males; 7.48 ± 4.22 years from symptoms onset). At 6 months, 30 patients completed the follow-up (90.9%). The NMSS total score was reduced by 27.3% (from 71.67 ± 37.12 at V0 to 52.1 ± 34.76 at V2; Cohen’s effect size = −0.97; *p* = 0.002). By domains, improvement was observed in sleep/fatigue (−40.1%; *p* < 0.0001), mood/apathy (−46.6%; *p* = 0.001), gastrointestinal symptoms (−20.7%; *p* = 0.029), and miscellaneous (−44.94%; *p* = 0.021). QoL also improved with a 18.4% reduction in the 39-item Parkinson’s Disease Quality of Life Questionnaire Summary Index (from 26.67 ± 17.61 at V0 to 21.75 ± 14.9 at V2; *p* = 0.001). A total of 13 adverse events in 11 patients (33.3%) were reported, 1 of which was severe (not related to opicapone). Dyskinesias and nausea were the most frequent (6.1%). Opicapone is well tolerated and improves global NMS burden and QoL in PD patients at 6 months.

## 1. Introduction

Parkinson’s disease (PD), the second most common neurodegenerative disease after Alzheimer’s disease, is a progressive neurodegenerative disorder causing motor and non-motor symptoms (NMS) that result in disability, loss of patient autonomy and caregiver burden [1]. The understanding of PD has changed over recent years, with the disease currently considered to be a neurodegenerative disorder involving a diversity of pathways and neurotransmitters that may explain, in part, the wide range of NMS that patients may have such as depression, anxiety, pain, cognitive impairment, apathy, gastrointestinal, urinary or cardiovascular symptoms, fatigue, or sleep problems [2,3]. NMS are frequent, disabling, and impact negatively on the quality of life (QoL) of PD patients [4], and strategies designed to improve NMS are necessary [5]. In clinical practice, the identification of NMS by the neurologist is very important, as well as knowing how they affect the patient. The NMS burden can be high even in the early stages of PD and impact patients’ QoL [6]. Moreover, NMS burden progresses over time in PD [7] and, very importantly, it is strongly correlated to motor complications [8]. In this context, a decrease of daily OFF episodes could help to improve some NMS in PD patients and a drug with an only dopaminergic effect (e.g., COMT inhibitor) could improve NMS [9,10,11] or hypothetically even the global NMS burden in PD patients [8].

Opicapone is a novel, long-acting, peripherally selective, once-daily, third generation COMT inhibitor [12]. Up to 32 trials have been conducted on opicapone use (with >900 subjects exposed to opicapone) [13]. In two Phase III clinical trials, opicapone 50 mg demonstrated to be superior to placebos in OFF-time reduction without increasing ON-time with troublesome dyskinesias in PD patients with moderate end-of-dose motor fluctuations [14,15]. Opicapone at doses of 5–50 mg/day has been found to be safe and well tolerated. The most common reported adverse effect was dyskinesia (16–21%) [14,15,16,17]. Regarding NMS, the effect of opicapone over them is not clear. The efficacy of opicapone on NMS was explored analyzing the data of the Non-Motor Symptoms Scale (NMSS) in PD patients who were included in the BIPARK II study, both in the double-blind and open-label phases [12,18]. At the end the double-blind phase, the NMSS total score slightly improved for both opicapone and placebo groups without significant differences between them. Numerical differences in favor of opicapone was seen for the sleep/fatigue domain. At the end of the open-label phase, a mean improvement of −4.2 points in the NMSS total score was still observed, and no worsening of any particular domain was detected [18]. More recently, a significant mean reduction of 6.8  ±  19.7 points (*p* < 0.0001) in the NMSS in 393 out of 495 PD patients treated with opicapone was observed at 3 months in a prospective, open-label, single-arm trial conducted in Germany and the United Kingdom under clinical practice conditions (OPTIPARK study) [11].

The aim of the present prospective open-label single-arm study (**OPEN-PD**, **Op**icapone **E**ffectiveness on **N**on-motor symptoms in **P**arkinson’s **D**isease) was to analyze the effectiveness of opicapone on global NMS burden (defined as the NMSS total score) in PD patients. Secondary objectives were to analyze the effectiveness of opicapone on the NMS burden of each domain of the NMSS, and also specifically on sleep, apathy, pain, health-related QoL, and autonomy for activities of daily living (ADL).

## 2. Material and Methods

OPEN-PD, Opicapone Effectiveness on Non-motor symptoms in Parkinson’s Disease, is a multicenter, observational (phase IV), prospective, open-label, follow-up study conducted in 5 centers from Spain. A total of 40 PD patients were expected to be included in the study. Inclusion criteria were: (1) diagnosis of PD according to the United Kingdom Parkinson’s Disease Society Brain Bank criteria [19]; (2) to be under levodopa therapy and have indication for receiving opicapone according to the neurologist criteria in his/her clinical practice; (3) NMSS total score at baseline >40; (4) age > 30 years old; (5) voluntary participation and signed informed consent form. Exclusion criteria were: (1) to be taking opicapone at the inclusion evaluation moment or to have been taking opicapone before; (2) to be under other COMT inhibitor therapy (entacapone or tolcapone) at the inclusion evaluation moment or to have received it in the previous month; (3) any contraindication to be treated with opicapone according to product data; (4) incapacity to complete the questionnaires adequately; (5) other disabling concomitant neurological disease (stroke, severe head trauma, neurodegenerative disease, etc.); (6) other severe and disabling concomitant non-neurological disease (oncological, autoimmune, etc.); (7) expected impossibility of long-term follow-up; (8) to be participating in a clinical trial and/or other type of study. All the neurologists who participated in the study of each center were experts on PD/movement disorders.

The study visits included (1) V0 (baseline); (2) V1 (2 months ± 14 days); and (3) V2 (6 months ± 30 days, end of the Observational Period). At baseline (V0), subjects completed an assessment that included motor symptoms (Hoehn and Yahr [H&Y] [20]; Unified Parkinson’s Disease Rating Scale [UPDRS] part III and part IV [21]; Freezing of Gait Questionnaire [FOGQ] [22]), NMS (NMSS [23]; Parkinson’s Disease Sleep Scale [PDSS] [24]; Apathy Scale (AS) [25]; King’s Parkinson’s Disease Pain Scale (KPPS) [26]; VAS-PAIN [27]), disability (Schwab & England Activities of Daily Living Scale [ADLS] [28]), and health related QoL (the 39-item Parkinson’s Disease Questionnaire [PDQ-39] [29]). The same assessment was performed at V1 and V2 except for H&Y and UPDRS-III (only at baseline). Moreover, Patient Global Impression of Change (PGIC) [30] was conducted at V1 and V2. Information on sociodemographic aspects, factors related to PD, comorbidity, and treatment was collected.

The primary objective was to analyze the effectiveness of opicapone on global NMS burden (defined as the NMSS total score) at V2 (6 months ± 30 days). The NMSS includes 30 items, each with a different non-motor symptom. The symptoms refer to the 4 weeks prior to assessment. The total score for each item is the result of multiplying the frequency (0, never; 1, rarely; 2, often; 3, frequent; 4, very often) x severity (1, mild; 2, moderate; 3, severe) and will vary from 0 to 12 points. The scale score ranges from 0 to 360 points. The items are grouped into 9 different domains: (1) Cardiovascular (items 1 and 2; score, 0 to 24); (2) Sleep/fatigue (items 3, 4, 5 and 6; score, 0 to 48); (3) Mood/apathy (items 7, 8, 9, 10, 11 and 12; score, 0 to 72); (4) Perceptual problems/hallucinations (items 13, 14 and 15; score, 0 to 36); (5) Attention/memory (items 16, 17 and 18; score, 0 to 36); (6) Gastrointestinal symptoms (items 19, 20 and 21; score 0 to 36); (7) Urinary symptoms (items 22, 23 and 24; score, 0 to 36); (8) Sexual dysfunction (items 25 and 26; score 0 to 24); (9) Miscellaneous (items 27, 28, 29 and 30; score, 0 to 48). Secondary objectives included: (1) to analyze the effectiveness of opicapone on NMS burden of each domain of the NMSS, and specifically also on sleep (PDSS), apathy (AS), and pain (KPPS); (2) to analyze the effectiveness of opicapone on motor complications (UPDRS-IV) and gait problems (FOGQ) including freezing of gait (FOG) (FOGQ-item 3); (3) to analyze the effectiveness of opicapone on health related QoL (PDQ-39) and functional capacity for ADL (ADLS); (4) to assess the clinical global impression of change according to the patient (PGIC); and (5) to analyze the safety and security of opicapone in PD patients.

Opicapone was administered as a once-daily 50 mg capsule. This study did not contemplate the switching of entacapone or tolcapone (COMT inhibitors) to opicapone. So, patients with PD who were being treated with another COMT inhibitor different from opicapone should take at least 1 month without taking a COMT inhibitor (entacapone and/or tolcapone) to be considered a candidate to participate in the study. During follow-up, any other medications different from opicapone should not have been modified (regimen, doses, etc.) except if the neurologist considered these changes absolutely necessary. All the changes including PD and not-PD related medications and levodopa-equivalent daily dose (LEDD) [31] of levodopa were recorded.

### 2.1. Data Analysis

Data were processed using SPSS 20.0 for Windows. Continuous variables were expressed as the mean ± SD or median and quartiles. Relationships between variables were evaluated using the Student’s *t*-test, the Mann–Whitney U test, or Spearman’s or Pearson’s correlation coefficient as appropriate (distribution for variables was verified by one-sample Kolmogorov–Smirnov tests). NMS burden was defined as: mild (NMSS 1–20); moderate (NMSS 21–40); severe (NMSS 41–70); and very severe (NMSS > 70) [32]. The PDQ-39 was expressed as a summary index (PDQ-39SI): (score/156) × 100. Each domain of the NMSS and PDQ-39 was expressed as a percentage: (score/total score) × 100.

The primary efficacy outcome was the change from baseline (V0) to the end of the observational period (6 months; V2) in the NMSS total score. The change from V0 to V2 in NMSS domains, PDSS, AS, KPPS, VAS-PAIN, UPDRS-IV, FOGQ, PDQ-39SI, and ADLS were the secondary efficacy outcome variables. Analyses on efficacy variables were performed with the ITT data set (all subjects who receive at least one pill of opicapone and had a baseline and treatment observation for the primary efficacy outcome measure). A paired-sample *t*-test or Wilcoxon’s rank sum test, as appropriate, was performed for testing the change from baseline. Cohen’s d formula was applied for measuring the effect size. The following was considered: <0.2—Negligible; 0.2–0.49—Small; 0.50–0.79—Moderate; and ≥0.80—Large. McNemar’s or marginal homogeneity tests were applied for comparing the frequency distribution of groups between V0 and V2. Values of *p* < 0.05 were considered significant.

The safety data set consists of all subjects for whom the study device was initiated. Safety analyses was assessed by adverse events (AEs). All AEs was coded using the current version of the Medical Dictionary for Regulatory Activities (MedDRA). The number and percentage of subjects with treatment emergent AEs by MedDRA system organ class and preferred term, by severity, and by relationship to study treatment as assessed by the investigator, was provided for overall subjects.

### 2.2. Standard Protocol Approvals, Registrations, and Patient Consents

For this study, we received approval from the *Comité de Ética de la Investigación Clínica de Galicia* from Spain (2017/475; 31/OCT/2017). Written informed consents from all participants in this study were obtained before the start of the study. OPEN-PD was classified by the AEMPS (*Agencia Española del Medicamento y Productos Sanitarios*) as a Post-authorization Prospective Follow-up study with the code DSG-OPI-2017-01.

### 2.3. Data Availability

The protocol and the statistical analysis plan are available on request. De-identified participant data are not available for legal and ethical reasons.

## 3. Results

A total of 33 out of 35 PD patients were included between July/2019 and June/2021 (age 63.3 ± 7.91; 60.6% males). Two patients selected finally refused to participate by their own decision. Data about sociodemographic aspects, comorbidities, antiparkinsonian drugs, and other therapies are shown in Table 1. The mean time from symptoms onset of PD was 7.48 ± 4.22 years. All patients were receiving oral levodopa, and none were under a second line therapy (pump infusion or deep brain stimulation). About two out of three patients were receiving a MAO-B inhibitor and/or a dopamine agonist and less than 10% amantadine or an anticholinergic agent. Benzodiazepines, antidepressant agents, and analgesic drugs were taken by 36.4%, 24.2%, and 21.2% of the patients, respectively. None were taking an antipsychotic agent. The mean LEDD was 820.89 ± 323.31 mg (range from 350 to 1812 mg).

At baseline (V0), 97% (32/33) of the patients presented with motor fluctuations and 42.4% (14/33) with dyskinesia. The mean UPDRS-III during the ON state was 21.61 ± 13.17. With regard to the NMS, the mean NMSS total score at baseline was 71.67 ± 37.12, presenting 69.7% (23/33) and 30.3% (10/33) of the patients with severe and with very severe NMS burden, respectively. Considering the different domains from the NMSS, the highest scores were in domains 2 (sleep/fatigue), 7 (urinary symptoms), and 3 (mood/apathy) (Table 2). Regarding QoL, the most affected domains were 3 (emotional well-being), 8 (bodily discomfort), and 6 (cognition) (Table 2).

At 6 months, 30 patients completed the follow-up (90.9%). The NMSS total score was reduced by 27.3% (from 71.67 ± 37.12 at V1 to 52.1 ± 34.76 at V2; Cohen’s effect size = -0.97; *p*=0.002) (Table 2). Six out of 30 patients presented a higher NMSS total score at the end of the follow-up compared to baseline (20%), whereas in the rest of the patients the NMSS total score was lower (range of decrease from 7 to 83 points). Compared to the score at V0, the change at V1 was significant too (*p* = 0.001) (Figure 1). By domains, improvement was observed in sleep/fatigue (−40.1%; Cohen’s effect size = −1.06; *p* < 0.0001), mood/apathy (−46.6%; Cohen’s effect size = −0.87; *p* = 0.001), gastrointestinal symptoms (-20.7%; Cohen’s effect size = −0.56; *p* = 0.029), and miscellaneous (−44.94%; Cohen’s effect size = −0.65; *p* = 0.021) (Table 2 and Figure 2). Compared to baseline, a significant improvement in sleep/fatigue and mood/apathy was reported at V1 whereas no differences were detected in any domain between V1 and V2 (Appendix A). At the end of the follow-up, the NMS burden by groups was 9.7% mild; 38.7% moderate; 25.8% severe; and 25.8% very severe (comparison to V1, *p* = 0.001) (Figure 3). Regarding other scales assessing NMS, no significant results were observed, although a trend of significance was detected for the KPPS (*p* = 0.075) (Table 2). A significant improvement was detected in item 27 of the NMSS (pain not explained for other known condition), changing the score from 4.15 ± 4.5 at V0 to 1.9 ± 3.39 at V2 (*p* = 0.007). With regard to motor symptoms, it was detected a significant reduction in the FOGQ (from 7.42 ± 5.62 at V0 to 6.03 ± 5.41 at V2; Cohen’s effect size = −0.56; *p* = 0.018). FOG at baseline was reported by 48.5% of the patients and 43.3% at the end of the follow-up (*p* = 0.687).

QoL also improved at V2 with a 18.4% reduction in the PDQ-39SI (from 26.67 ± 17.61 at V0 to 21.75 ± 14.9 at V2; Cohen’s effect size = -0.99; *p* = 0.001) compared to the score at baseline. Specifically, by domains, the difference between V0 and V2 was significant for PDQ-39SI-3 (Emotional well-being) (*p* = 0.004), PDQ-39SI-4 (Stigmatization) (*p* = 0.009), and PDQ-39SI-8 (Pain and discomfort) (*p* = 0.023) (Table 2). At 6 months, 17 patients out of 30 (56.7%) felt better regarding the PGIC: 1 very much improved; 9 much improved; 7 minimally improved; 10 had no changes; and 3 were minimally worse.

A total of 13 adverse events in 11 patients (33.3%) were reported, 1 of which was severe (not related to opicapone) (Table 3). Dyskinesias and nausea were the most frequent (6.1%). Two patients discontinued due to an adverse event related to opicapone (nausea and insomnia), whereas in the third case it was a personal decision of the patient due to a lack of effect of the drug.

## 4. Discussion

The present study observed that global NMS burden (NMSS total score) improved in PD patients 6 months after starting with opicapone. Specifically, an improvement was detected in domains of the NMSS related to sleep, fatigue, mood, gastrointestinal symptoms, and pain. Moreover, the effect was significant at 2 months after starting with opicapone, and an improvement in QoL was observed as well. This is the first prospective study specifically designed for assessing the change in global NMS burden in PD patients after been treated with this drug.

NMS are frequent in PD, and their recognition is very important because of their negative impact on QoL [4,32]. In this study, PD patients had to present with severe or very severe NMS burden (NMSS total score > 40) at baseline for being included. Recently, it was observed that up to 30.5% and 44.9% of PD patients with a H&Y stage of 1 and 2, respectively, had severe or very severe NMS burden, and, importantly, patients with a lower H&Y stage may be more affected if they had a greater NMS burden than others with a higher H&Y stage [6]. Therefore, strategies designed to improve NMS are necessary [5]. In this context and considering that some NMS can be related to the deficit of other neurotransmitters different than dopamine (e.g., depression and serotonin), a key question is if NMS can improve with a drug with an only dopaminergic effect [33]. Increasing dopamine activity not only in the striatum but also in other areas of the brain could improve some NMS such as attention and executive functions, depression, anxiety, apathy, restless legs and periodic limb movements, urinary urgency, nocturia, dribbling of saliva, constipation, pain, or fatigue [33,34,35,36,37]. Specifically, some studies with first and second generation COMT inhibitors (entacapone and tolcapone) observed a benefit by the patients on some NMS [9,10,38,39], but really the evidence is scarce, possibly in part due to the fact that NMS are an emerging topic and have been much more studied in recent years. Moreover, if COMT inhibitors such as opicapone can improve ON time, non-motor fluctuations [40], NMS related to OFF episodes [41] and global NMS burden as a whole [8] could improve as well.

Opicapone is a third generation COMT inhibitor rationally designed to reduce the risk of toxicity and improve COMT inhibitory potency and peripheral tissue selectivity compared with other COMT inhibitors [42]. The efficacy and safety of opicapone in reducing OFF time in patients with PD and established motor fluctuations has been well established in three randomized, double-blind, placebo-controlled trials (BIPARK I; BIPARK II; COMFORT-PD) [14,15,43] and observational studies [11,44,45,46]. However, the effect of opicapone on NMS is a relatively unexplored aspect [42]. In the BIPARK II study, NMS were assessed with the NMSS at different time points, including baseline, the end of the double-blind phase, and the end of the open-label phase. At the end of the double-blind period, NMSS scores slightly improved across the opicapone and placebo groups, with no significant differences between them (placebo, −5.2; opicapone 25 mg, −2.0; opicapone 50 mg, −4.9) [16]. At the 1-year open-label endpoint, a mean improvement of −4.2 in NMSS total score was still maintained [16]. However, data about domains of the NMSS and even the NMSS total score at baseline was not provided in all groups. In those patients receiving opicapone 50 mg (N = 325), the mean baseline NMSS total score was 37.9 ± 28.7. In our study, the score was very much higher (71.67 ± 37.12) due to the inclusion criteria and aim proposed, which is important because the probability of having an improvement is related to the baseline score of the scale [47]. Similar results have previously been observed in the BIPARK I study (−5.7, placebo; −2, opicapone 50 mg). In the third pivotal study of opicapone conducted in Japan, NMS were not assessed [43]. In the only other published study analyzing how NMS changed in PD patients treated with opicapone, the OPTIPARK study [11], a decrease in the NMSS total score of 6.8  ±  19.7 points (*p* < 0.0001) at 3 months was observed. This was a prospective, open-label, single-arm trial conducted in Germany and the United Kingdom under clinical practice conditions, and data about NMS were collected in 393 PD patients. Again, the NMSS total score at baseline was lower than in our study (44.6 ± 30.3), and the change from baseline to the visit at 3 months in the NMS burden was not the primary efficacy outcome. In this study [11], Reichmann et al. reported a significant improvement in all domains of the NMSS except in domain 4 (perceptual problems/hallucinations), but the size effect was not calculated, and it is not clear whether it can be considered clinically relevant, with a decrease in the score that varied from 13.3% (cardiovascular symptoms; -0.2 from 1.5 ± 2.38 at baseline; *p* = 0.0310) to 22.4% (mood/apathy; −1.5 ± 6.82 from 6.7 ± 9.8; *p* < 0.0001). In our study, the greatest improvement was observed in the mood/apathy and sleep/fatigue domains, both with large effect. In the BIPARK II study, a significant signal was seen for the sleep/fatigue domain where the 50 mg dose reduced the NMSS sleep/fatigue score by −1.2 points versus −0.5 points with a placebo [42]. The small sample size of our cohort could explain why the change in sleep (PDSS) and pain (KPPS) scales was not significant. The PDSS was used in the BIPARK II study, but differences were not detected compared to a placebo [16]. In this line of research, studies such as OASIS (OpicApone Sleep dISorder; EudraCT number 2020–001176-15) and OCEAN (the OpiCapone Effect on motor fluctuations and pAiN; EudraCT number 2020–001175-32) are currently underway to evaluate the effect of opicapone 50 mg on sleep and pain, respectively. On the other hand, our study is the first prospective one exploring the effect of opicapone on motivation/apathy using a specific validated scale, but we did not find differences. A recent publication that reviewed data of small retrospective series of PD patients treated with opicapone in Spain suggests a possible positive effect of opicapone on NMS after 6 to 12 months, especially on sleep [48]. However, and in agreement with our findings, the frequency of apathy in 60 PD patients treated with opicapone in real clinical practice did not change at 6 months (32%) and 12 months (33%) compared to the baseline visit (32%) [48]. More studies designed to evaluate the effect of opicapone over NMS are really needed.

In addition to NMS, we observed in our study a trend of significant improvement in motor complications (UPDRS-IV; *p* = 0.083) and a significant improvement in gait problems (*p* = 0.018). Although the frequency of patients reporting FOG was similar before and after treatment with opicapone, gait problems as a whole may improve due to motor signs improvement and OFF time reduction [49]. Only two small studies looked at the effects of COMT inhibitors on gait parameters, providing support for tolcapone as an effective add-on to levodopa to prolong beneficial effects on gait speed [50,51]. In previously published studies of opicapone, its effect on gait was not analyzed. However, as we detected here, QoL improved significantly in PD patients treated with opicapone in real clinical practice [11]. We used the PDQ-39 and observed improvement in emotional well-being, stigmatization, and pain and discomfort. On the contrary, in the OPTIPARK study the brief version (PDQ-8) was used, and data about domains was not provided in the publication [11]. Contrary to this study, we did not find improvement in the autonomy for ADL.

Opicapone was not only effective but also safe and well-tolerated, with a very high drug maintenance rate at 6 months, above 90%. The rate was 79.4% (393/495) at 3 months and 85.3% (81/95) at 6 months for all the cohort and for the United Kingdom subgroup only, respectively, from the real clinical practice OPTIPARK cohort [11], and 92.2% (107/116) and 83.1% (128/154) in the double-blind phase of BIPARK I and BIPAK II studies, respectively [14,15]. The results about adverse events are in line with other studies [11,13,14,15,16,17,18,43,44,45,46], even with a lower percentage of events reported. Dyskinesia, as in some studies, was the most frequent adverse event in our study. The European public assessment report (EPAR) for opicapone states that dyskinesias were reported in more than 10% of participants receiving opicapone, in which case it may be necessary to reduce the levodopa dose within the first days to first weeks after starting opicapone to prevent severe dyskinesias [44]. This good tolerability of the drug was accompanied by an improvement according to the PGIC in almost 60% of the cases, in line with other reports [43,48].

Our study has some important limitations. The most important one is related to the study design itself, and since there is not a comparative arm with placebo, the results should be interpreted with caution. Second, the sample size is small, and it is possible that the changes observed in some variables are not significant due to this. In fact, due to different problems (i.e., administrative, commercial distribution of the drug in Spain, COVID-19 pandemic), the study was closed before reaching the initially planned sample size (N = 40). Third, the effect of opicapone on NMS was analyzed in PD patients with a severe or very severe NMS burden (NMSS total score > 40); therefore, the results cannot be extrapolated to patients with a mild or moderate NMS burden (NMSS total score ≤ 40). Fourth, mood was not assessed with a specific scale. On the other hand, this is the first study designed to assess the effect of opicapone on NMS burden in PD patients and the first one in which changes in some NMS such as pain, apathy, or sleep have been exhaustively analyzed. Despite some limitations, the results are novel and of great interest because there is a lack of knowledge about what benefits can opicapone produce over many symptoms in PD patients.

## 5. Conclusions

In conclusion, opicapone is well tolerated and improves global NMS burden and QoL in PD patients. Well-designed studies are necessary to analyze in detail the possible beneficial effect of opicapone on NMS in patients with PD.

## Figures and Tables

**Figure 1 brainsci-12-00383-f001:**
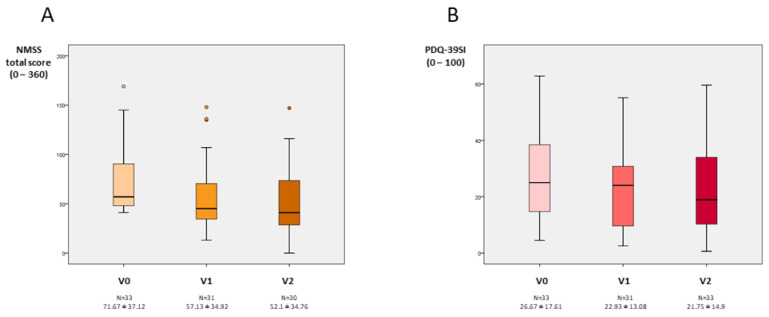
(**A**). NMSS total score at V0 (baseline), V1 (2 months ± 15 days), and V2 (6 months ± 30 days). V2 vs. V0, *p* = 0.002; V1 vs. V0, *p* = 0.001; and V2 vs. V1, *p* = 0.202. (**B**). PDQ-39SI at V0, V1, and V2. V2 vs. V0, *p* = 0.001; V1 vs. V0, *p* = 0.008; and V2 vs. V1, *p* = 0.496. Data are presented as box plots, with the box representing the median and the two middle quartiles (25–75%). *p* values were computed using the Wilcoxon signed-rank test. Mild outliers (O) are data points that are more extreme than Q1—1.5. NMS, non-motor symptoms; PDQ-39SI, 39-item Parkinson’s Disease Questionnaire Summary Index.

**Figure 2 brainsci-12-00383-f002:**
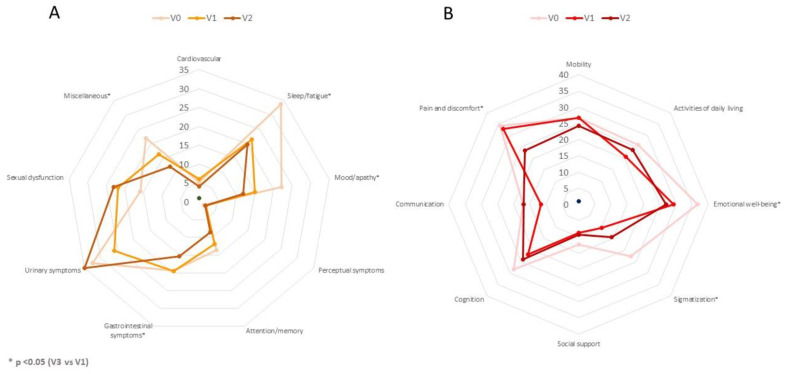
(**A**). Mean score on each domain of the NMSS scale at V0 (baseline), V1 (2 months ± 15 days), and V2 (6 months ± 30 days). The difference between V2 and V0 was significant for NMSS-2 (Sleep/fatigue) (*p* < 0.0001), NMSS-3 (Mood/apathy) (*p* < 0.001), NMSS-6 (Gastrointestinal symptoms) (*p* = 0.029), and NMSS-9 (Miscellaneous) (*p* = 0.021). (**B**). Mean score on each domain of the PDQ-39SI at V0, V1, and V2. The difference between V2 and V0 was significant for PDQ-39SI-3 (Emotional well-being) (*p* = 0.004), PDQ-39SI-4 (Stigmatization) (*p* = 0.009), and PDQ-39SI-8 (Pain and discomfort) (*p* = 0.023). *p* values were computed using the Wilcoxon signed-rank test.

**Figure 3 brainsci-12-00383-f003:**
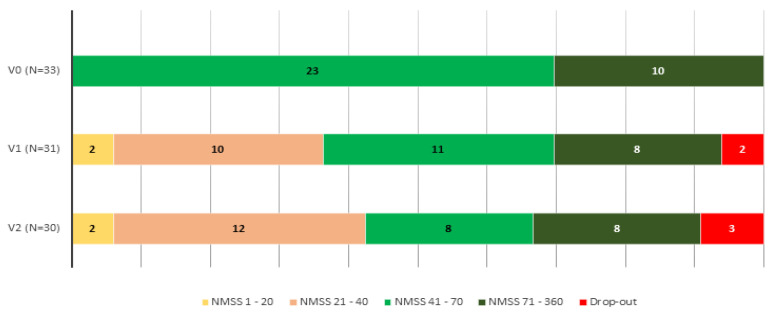
NMS burden with regard to the NMSS total score (0–20, slight burden; 21–40, moderate burden; 41–70, severe burden; 71–360 very severe burden) at V0, (baseline), V1 (2 months ± 14 days), and V2 (6 months ± 30 days). V2 vs. V0, *p* = 0.001; V1 vs. V0, *p* = 0.001; V2 vs. V1, *p* = 0.366.

**Table 1 brainsci-12-00383-t001:** Data about sociodemographic aspects, comorbidities, antiparkinsonian drugs, and other therapies at baseline (N = 33).

Age	63.3 ± 7.91 (48–77)	Time from symptoms onset	7.48 ± 4.22 (2–20)
Gender (males) (%)	60.6		
Ethnicity (%)		Motor fluctuations (%)	97
- Caucasian	97	Dyskinesia (%)	42.4
- Other	3		
		Treatment for PD (%):	
Civil status (%):		- Levodopa	100
- Married	78.8	- MAO-B inhibitor	63.6
- Widowed	3	- Dopamine agonists:	66.7
- Single	6.1	* Pramipexole	33.3
- Divorced	9.1	* Ropinirole	12.1
- Other	3	* Rotigotine	21.2
		- Amantadine	9.1
Living style (%)		- Anticholinergic drug	3
- With the partner	75.8		
- With another family member	6.1	L-dopa daily dose (mg)	648.46 ± 372.44 (200–1792)
- With a son/daughter	6.1	DA daily dose (mg)	162.84 ± 161.03 (0–630)
- Other	12	LEDD (mg)	820.89 ± 323.31 (350–1812)
Habitat (%):		Other treatments (%):	
- Rural (<5.000)	21.2	- Antidepressant	24.2
- Semiurban (5.000–20.000)	24.2	- Benzodiazepine	36.4
- Urban (>20.000)	54.6	- Antipsychotic	0
		- Analgesic	21.2
Comorbidities (%):			
- Arterial hypertension	39.4	Number of anti-PD drugs	2.75 ± 1.14 (1–5)
- Diabetes mellitus	15.2	Number of non-PD drugs	2.51 ± 2.56 (0–9)
- Dyslipemia	27.3	Total number of drugs	5.24 ± 2.92 (1–11)
- Hiperuricemia	3	Number of pills for PD	5.82 ± 1.6 (3–9)
- Cardiomyopathy	6	N. of pills for other cause	2.54 ± 2.69 (0–9.5)
- Cardiac arrhythmia	12.1	Total number of pills	8.37 ± 3.24 (3–18.5)
- Smoking	9.1		
- Alcohol consumption	18.2		

The results represent % or mean ± SD (range).

**Table 2 brainsci-12-00383-t002:** Change in the score of the NMSS and other scales of the study from V0 (baseline; N = 33) to V2 (6 months ± 30 days; N = 30).

	V0	V2	Cohen’s d	∆ V0–V2	*p*
**MOTOR ASSESSMENT**					
**H&Y-OFF**	2.5 [2, 3]	N. A.	N. A.	N. A.	N. A.
**H&Y-ON**	2 [1.5, 2]	N. A.	N. A.	N. A.	N. A.
**UPDRS-III-ON**	21.61 ± 13.17	N. A.	N. A.	N. A.	N. A.
**UPDRS-IV**	4.48 ± 2.09	3.87 ± 2.5	−0.38	−13.6%	0.083
**FOGQ**	7.42 ± 5.62	6.03 ± 5.41	−0.56	−18.7%	0.018
**NON MOTOR ASSESSMENT**					
**NMSS total score**	71.67 ± 37.12	52.1 ± 34.76	−0.97	−27.3%	0.002
- Cardiovascular	5.3 ± 9.21	4.03 ± 6.03	−0.24	−23.9%	0.346
- Sleep/fatigue	33.08 ± 19.02	19.82 ± 16.4	−1.06	−40.1%	<0.0001
- Mood/apathy	22.22 ± 22.58	11.87 ± 14.82	−0.87	−46.6%	0.001
- Perceptual symptoms	1.59 ± 4.76	1.88 ± 4.5	+0.12	+18.2%	0.334
- Attention/memory	13.55 ± 15.8	8.6 ± 19.62	−0.32	−36.5%	0.091
- Gastrointestinal symptoms	19.44 ± 16.27	15.41 ± 16.36	−0.56	−20.7%	0.029
- Urinary symptoms	32.57 ± 26.01	34.49 ± 26.02	+0.09	+5.8%	0.726
- Sexual dysfunction	15.78 ± 26.2	22.98 ± 30.97	+0.43	+45.6%	0.099
- Miscellaneous	21.96 ± 18.56	12.09 ± 14.11	−0.65	−44.94%	0.021
**PDSS**	104.62 ± 22.51	108.48 ± 26.86	+0.24	+3.6%	0.267
**AS**	13.76 ± 8.4	14.6 ± 8.71	+0.11	+6.1%	0.801
**KPPS**	14.33 ± 13.5	10.47 ± 9.62	−0.44	−26.9%	0.075
- Musculoskeletal pain	4.06 ± 3.46	3.47 ± 3.53	−0.14	−14.5%	0.474
- Chronic pain	1.76 ± 3.51	0.63 ± 1.75	−0.12	−64.2%	0.073
- Fluctuation-related pain	2.7 ± 4.24	1.23 ± 2.55	−0.06	−54.4%	0.133
- Nocturnal pain	4.18 ± 6.44	2.93 ± 4.2	−0.34	−29.9%	0.266
- Oro-facial pain	0.52 ± 1.46	0.43 ± 1.61	−0.09	−17.3%	0.524
- Discoloration, edema/swelling	0.18 ± 0.72	0.53 ± 1.47	+0.38	+194.4%	0.109
- Radicular pain	0.94 ± 2.35	1.23 ± 2.73	+0.09	+30.8%	0.878
**VAS-PAIN**	4.09 ± 3.11	4.55 ± 2.5	+0.41	+11.2%	0.187
**QOL AND AUTONOMY**					
**PDQ-39SI**	26.67 ± 17.61	21.75 ± 14.9	−0.99	−18.4%	0.001
- Mobility	26.74 ± 19.62	24.16 ± 24.68	−0.45	−9.6%	0.064
- Activities of daily living	26.01 ± 20.46	23.61 ± 19.82	−0.33	−8.9%	0.130
- Emotional well-being	36.74 ± 26.95	26.94 ± 21.49	−0.83	−26.7%	0.004
- Stigmatization	22.72 ± 27.41	14.37 ± 20.77	−0.75	−36.7%	0.009
- Social support	12.37 ± 18.76	9.44 ± 15.43	−0.39	−23.6%	0.244
- Cognition	28.41 ± 19.45	24.16 ± 25.2	−0.25	−14.9%	0.306
- Communication	17.17 ± 20.61	16.94 ± 18.63	−0.09	−1.3%	0.895
- Pain and discomfort	34.34 ± 21.52	23.33 ± 19.98	−0.67	−32.1%	0.023
**ADLS**	80 ± 13.91	82.33 ± 14.54	−0.35	−2.9%	0.197

*p* values were computed using the Wilcoxon signed-rank test. The results represent mean ± SD or median [p25, p75]. Domains of the NMSS and PDQ-39SI were expressed as a percentage to be able to establish comparisons on their severity between them. Cohen’s d formula was applied for measuring the effect size. It was considered: small effect = 0.2; medium effect = 0.5; large effect = 0.8. N. A., not applicable. ADLS, Schwab & England Activities of Daily Living Scale; FOGQ, Freezing of Gait Questionnaire, H&Y: Hoenh & Yahr; KPPS, King’s PD Pain Scale; NMSS, Non-Motor Symptoms Scale; PDQ-39SI, 39-item Parkinson’s Disease Quality of Life Questionnaire Summary Index; PDSS, Parkinson’s disease Rating Scale; UPDRS, Unified Parkinson’s Disease Rating Scale; VAS-Pain, Visual Analog Scale-Pain.

**Table 3 brainsci-12-00383-t003:** Adverse events in patients from V0 to V2.

	N
Total AEs, N	13
- Dyskinesia	2
- Nausea	2
- Unrest	1
- Visual hallucinations	1
- Insomina	1
- Vivid dreams	1
- Tiredness	1
- Insomnia	1
- OFF time increase	1
- Arthritis in both wrists	1
- Supraspinatus tendonitis	1
Patients with at least one AE, N (%)	11 (33.3)
At least possibly related AEs, N	8
Patients with at least possibly* related to opicapone AEs, N (%)	7 (21.2)
Total SAEs, N	1
- Arthritis in both wrists	
Patients with al least one SAE, N (%)	1 (3)
At least possibly * related to opicapone SAEs, N	0
Patients with at least possibly related to opicapone SAEs, N (%)	0 (0)
Patients with at least one AE leading to discontinuation, N (%)	2 (6.1)
Patients with at least one possibly* related to opicapone AE leading to discontinuation N (%)	2 (6.1)
Deaths, N (%)	0 (0)

* Considered “possibly”, “probably” or “definitely” related to treatment (opicapone). AE, adverse event; SAE, serious adverse event.

## Data Availability

The protocol and the statistical analysis plan are available on request. Deidentified participant data are not available for legal and ethical reasons.

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
