# Peer review of "Opicapone Improves Global Non-Motor Symptoms Burden in Parkinson’s Disease: An Open-Label Prospective Study"

_brainsci, 2022, doi:10.3390/brainsci12030383_

Round 1

Reviewer 1 Report

Thank you for the opportunity to review this interesting manuscript.

This multicenter, prospective, open label study aimed to evaluate the change in the total burden of non-motor symptoms (measured as the total score on the Non-motor Symptoms Scale, NMSS) between the baseline and after the 6-month treatment with Opicapone, a new generation COMT inhibitor. 30 patients with advanced PD completed the study. The NMSS total score was significantly reduced after only 2 months of the treatment, with only a few adverse events, and this effect persisted following 6 months of the treatment.

This is an interesting study with important clinical implications and the manuscript is very well written.

I have added some minor suggestions below:

Abstract: The word „some” is used twice in the same sentence. I would suggest to either re-formulate or remove the first sentence.

Hoehn and Yahr stage should be added to the demographic and disease-specific features.

Introduction: Line 47: “Large range” – I suggest “wide range”

Line 51-52: *the identification of the NMSS is very important

Methods:

I suggest using passive for the inclusion and exclusion criteria.

The baseline visit should be named V0 instead of V1.

Line 108: “Subjects completed an assessment” – some of the assessments are rater-completed.

Line 141-145: This should be mentioned earlier, under the inclusion/exclusion criteria.

Results:

Line 195: *taken instead of taking

Table 2: Please use the word ethnicity instead of race

Table 2 should also show the scores at the V2 – it is important to show that significant improvement is seen already following a 2-month treatment with Opicapone

Line 218: “In visit” is incorrect, please change to “at”

Discussion:

Line 279: “because of their negative impact”

Line 282: H&Y stage

Line 294: *possibly

Line 315: *which is important

Line 366: *with very high drug maintenance rate

Dyskinesia is mentioned as the most common adverse event. Maybe the authors could also mention that, in the clinical practice, this is usually managed by a reduction of the levodopa dose.

Author Response

A Coruña, 09.MAR.2022

Ms. Natalia Oprea

Assistant Editor

Brain Sciences

Dear Ms. Natalia Oprea,

We are submitting a revised version (R1) of the manuscript entitled “Opicapone Improves Global Non-Motor Symptoms Burden in Parkinson´s Disease: An Open-label Prospective Study”. 

Thank you for your letter of March 9, 2022. We appreciate all your comments and suggestions as well those from the reviewer's of our manuscript: we have modified the manuscript as their suggestion and a point-by-point response to the reviewers’ comments is enclosed. Moreover, we have corrected some errors in the edition (different size letter, alignment to the left in the tables, some title without bold, etc.). We would be pleased to provide additional information if requested or to further modify the text.

Correspondence will be send to the following address: Dr. Diego Santos-García, Department of Neurology – CHUAC (Complejo Hospitalario Universitario de A Coruña), c/ As Xubias 84, 15006, A Coruña, Spain; e-mail: diegosangar@yahoo.es.

We look forward to hearing from you. Thank you for your attention to our submission.

Sincerely yours,

Diego Santos García 

Reviewer 1

Thank you for the opportunity to review this interesting manuscript.

This multicenter, prospective, open label study aimed to evaluate the change in the total burden of non-motor symptoms (measured as the total score on the Non-motor Symptoms Scale, NMSS) between the baseline and after the 6-month treatment with Opicapone, a new generation COMT inhibitor. 30 patients with advanced PD completed the study. The NMSS total score was significantly reduced after only 2 months of the treatment, with only a few adverse events, and this effect persisted following 6 months of the treatment.

This is an interesting study with important clinical implications and the manuscript is very well written.

I have added some minor suggestions below:

Abstract: The word „some” is used twice in the same sentence. I would suggest to either re-formulate or remove the first sentence.

Hoehn and Yahr stage should be added to the demographic and disease-specific features.

Introduction: Line 47: “Large range” – I suggest “wide range”.

Line 51-52: *the identification of the NMSS is very important.

AUTHORS – Thank you very much for your comment. The changes have been made as you suggest.

Methods:

I suggest using passive for the inclusion and exclusion criteria.

The baseline visit should be named V0 instead of V1.

Line 108: “Subjects completed an assessment” – some of the assessments are rater-completed.

Line 141-145: This should be mentioned earlier, under the inclusion/exclusion criteria.

AUTHORS – Thank you very much for your comment. The changes have been made as you suggest.

Results:

Line 195: *taken instead of taking

Table 2: Please use the word ethnicity instead of race

Table 2 should also show the scores at the V2 – it is important to show that significant improvement is seen already following a 2-month treatment with Opicapone

Line 218: “In visit” is incorrect, please change to “at”

AUTHORS – Thank you very much for your corrections and improving the english style. The data about the NMS in all visits is provided in the Table 1 – Supplementary Material (p value of comparison between the three visits is provided).

Discussion:

Line 279: “because of their negative impact”

Line 282: H&Y stage

Line 294: *possibly

Line 315: *which is important

Line 366: *with very high drug maintenance rate

AUTHORS – Many thanks again for your comments. The changes have been made.

Dyskinesia is mentioned as the most common adverse event. Maybe the authors could also mention that, in the clinical practice, this is usually managed by a reduction of the levodopa dose.

AUTHORS – Many thanks for your comment. We have added a sentence about this aspect: “The European public assessment report (EPAR) for opicapone states that dyskinesias were reported in more than 10% of participants receiving opicapone in which case it may be necessary to reduce the levodopa dose within the first days to first weeks after starting opicapone to prevent severe dyskinesias [44]”.

Reviewer 2 Report

In this manuscript the authors present an open-label study of opicapone for non-motor symptoms of Parkinson's disease.  The impact of the study is limited solely by its design - previous more rigorous studies (randomized phase III trials) have not shown a benefit in non-motor symptoms with opicapone. The biggest strength of the study is the unique patient population of people who had severe non-motor symptoms. These symptoms are challenging to manage clinically, so there is great unmet need  in this area and this paper is still of value for that reason. This will be a good addition to the literature on opicapone. 

Author Response

A Coruña, 09.MAR.2022

Ms. Natalia Oprea

Assistant Editor

Brain Sciences

Dear Ms. Natalia Oprea,

We are submitting a revised version (R1) of the manuscript entitled “Opicapone Improves Global Non-Motor Symptoms Burden in Parkinson´s Disease: An Open-label Prospective Study”.

Thank you for your letter of March 9, 2022. We appreciate all your comments and suggestions as well those from the reviewer's of our manuscript: we have modified the manuscript as their suggestion and a point-by-point response to the reviewers’ comments is enclosed. Moreover, we have corrected some errors in the edition (different size letter, alignment to the left in the tables, some title without bold, etc.). We would be pleased to provide additional information if requested or to further modify the text.

Correspondence will be send to the following address: Dr. Diego Santos-García, Department of Neurology – CHUAC (Complejo Hospitalario Universitario de A Coruña), c/ As Xubias 84, 15006, A Coruña, Spain; e-mail: diegosangar@yahoo.es.

We look forward to hearing from you. Thank you for your attention to our submission.

Sincerely yours,

Diego Santos García

Reviewer 2.

In this manuscript the authors present an open-label study of opicapone for non-motor symptoms of Parkinson's disease.  The impact of the study is limited solely by its design - previous more rigorous studies (randomized phase III trials) have not shown a benefit in non-motor symptoms with opicapone. The biggest strength of the study is the unique patient population of people who had severe non-motor symptoms. These symptoms are challenging to manage clinically, so there is great unmet need  in this area and this paper is still of value for that reason. This will be a good addition to the literature on opicapone.

AUTHORS – Thank you very much for your comment. We greatly appreciate your feedback on this study.
